# Gas Chromatography Combustion Isotope Ratio Mass Spectrometry to Detect Differences in Four Compartments of Simmental Cows Fed on C3 and C4 Diets

**DOI:** 10.3390/molecules27072310

**Published:** 2022-04-02

**Authors:** Silvia Pianezze, Mirco Corazzin, Luana Bontempo, Angela Sepulcri, Elena Saccà, Matteo Perini, Edi Piasentier

**Affiliations:** 1Centro Trasferimento Tecnologico, Fondazione Edmund Mach, San Michele All’Adige, 38098 Trento, Italy; silvia.pianezze@fmach.it; 2Dipartimento di Scienze Agroalimentari, Ambientali e Animali, University of Udine, 33100 Udine, Italy; mirco.corazzin@uniud.it (M.C.); angela.sepulcri@uniud.it (A.S.); elena.sacca@uniud.it (E.S.); edi.piasentier@uniud.it (E.P.); 3Centro Ricerca e Innovazione, Fondazione Edmund Mach, San Michele All’Adige, 38098 Trento, Italy; luana.bontempo@fmach.it

**Keywords:** compound-specific gas chromatography, fatty acids, dietary regime, carbon stable isotope, animal metabolism

## Abstract

Fatty acids (FAs) metabolism in animals represents an important field of study since they influence the quality and the properties of the meat. The aim of this study is to assess the possibility to discriminate the diets of cows in different animal compartments and to study the fate of dietary FAs in the bovine organism, using carbon isotopic ratios. Five FAs, both essential (linoleic and linolenic) and non-essential (palmitic, stearic, and oleic) in four compartments (feed, rumen, liver, meat) of animals fed two different diets (based on either C3 or C4 plants) were considered. For all compartments, the carbon isotopic ratio (δ^13^C) of all FAs (with few exceptions) resulted significantly lower in cows fed on C3 than C4 plants, figuring as a powerful tool to discriminate between different diets. Moreover, chemical reactions taking place in each animal compartment result in fraction processes affecting the δ^13^C values. The δ^13^C_FAs_ tendentially increase from feed to meat in group C3. On the other hand, the δ^13^C_FAs_ generally increase from rumen to liver in group C4, while δ^13^C_FAs_ of rumen and meat are mostly not statistically different. Different trends in the δ^13^C_FAs_ of the two groups suggested different FAs fates depending on the diet.

## 1. Introduction

Fatty acids (FAs) play an important role in bovine digestive processes. Essential FAs, such as linoleic and linolenic, do not derive from endogenous synthesis, being only of dietary origin, while non-essential ones may be synthesized de novo in the organism [1].

In recent years, stable isotope ratio analysis (SIRA) has been used to investigate the authenticity of animal-derived food, including cheese [2], milk [3], butter [4], and meat [5]. Several works have been carried out with the aim to discriminate, through SIRA, between bovines [6,7,8], lambs [9], and sheep [10] fed on different feeding regimes. The cited works were carried out by focusing on bulk samples of different compartments. Carbon isotopic ratio (δ^13^C) was included in all the mentioned studies, as it is the most suitable to give information about the dietary regime of animals. Indeed, δ^13^C is mainly influenced by the photosynthetic pathway of the plants present in the animal diet [11]. It is well known that C4 plants (such as maize) have δ^13^C values between −14 and −12‰, while C3 plants (such as *Medicago Sativa* L. *grass*) values range between −30 and −23‰ [12,13,14]. Forages and concentrates included in the animal diet may derive from both C3 and C4 plants.

Compound-specific stable isotope analysis (CSIA), which makes it possible to collect the stable isotope values of individual organic compounds, such as FAs, offers potential solutions to some of the limitations of bulk samples containing isotopically distinct compounds [15]. In some cases, it was demonstrated that compound-specific analysis can provide more detailed information than bulk analysis [16]. In this sense, FAs analysis may provide a deeper understanding of the processes taking place in animal organisms and may also be a more precise dietary regime information supplier. Only a few works studied FAs to obtain information about animals’ diets [17,18,19]. Nevertheless, in all cited cases, all the diet components were classified as C3.

In this work, for the first time, the fate of FAs in the bovine organism was studied considering both C3 and C4 diets. Some preliminary data of the C3 experiment are reported in previous work [18]. Since the obtained results highlight different behaviors for the C3 and C4 groups, all the gathered data were considered, to compare the two diets and to evaluate the differences in the diet-dependent metabolic paths of FAs. For the first time, the δ^13^C of five FAs, both essentials (linoleic 18:2n-6 and linolenic 18:3n-3) and non-essentials (palmitic 16:0, stearic 18:0, oleic 18:1n-9), were collected from four different compartments, namely feed, rumen, liver, and meat, through CG-C-IRMS (gas chromatography-combustion-isotope ratio mass spectrometer). The aims of this work were: (1) to discriminate, by CSIA of the FAs in the four compartments, between cows fed with C3 and those fed with C4 plants, which can be useful for traceability purposes; (2) to obtain further information about the metabolic pathway of the FAs, which can be influenced by the diet, in the bovine organism.

## 2. Results and Discussion

The average pH of the rumen was 6.5 ± 0.05, similar for all experimental groups (*p* > 0.05) and all the animals showed a value within the normal range (6.2–7.2) [20], meaning that cows had normal ruminal fermentation balance.

As the compartment × diet interaction turned out to be statistically significant for all the variables considered (*p* < 0.05; data not reported), the main effects of compartment and diet could not be discussed separately. For this reason, the results of the interactions are reported in Figure 1, Figure 2 and Figure 3.

### 2.1. Fatty Acids Quantification and Carbon Isotope Ratio Analysis: Diet Effects

From an overall point of view, it is worth noting that both the quantification of the FAs and their δ^13^C made it possible to discriminate between the two different diets provided to the animals.

In the C4 group feed, the concentration of all the considered FAs was higher than in the C3 one (*p* < 0.05), with the only exception of C18:3n-3 (Figure 1). The differences observed in the feed FAs content were the same as those observed in meat (*p* < 0.05). Being essential, C18:2n-6 and C18:3n-3 have not been synthesized in ruminants and their concentration in the adipose tissues is thus closely related to their dietary amount and the proportional loss from the rumen [21]. The C4 group had a higher C18:2n-6 intake (feed compartment), but a lower C18:3n-3 one than the C3 group; however, focusing on the meat compartment, the C4 group had a higher content in both C18:2n-6 and C18:3n-3. Therefore, it could be hypothesized that the fraction of C18:3n-3 escaping the rumen was higher in the C4 group, probably because of a lower retention time and/or a different rumen biohydrogenation, as against the C3 one.

Figure 2, where the FAs content is expressed as a percentage (g/100 g of total FAs), shows that the C4 group feed had a higher content in C18:1n-9 and C18:2n-6, but a lower amount of C16:0, C18:0 and C18:3n-3 (*p* < 0.05; Figure 2) as against the C3 one. This agrees with the high content in C18:1n-9 and C18:2n-6 in corn [22], which represented the major component of the C4 group diet. The relative amount of C18:2n-6 (*p* < 0.05) and C18:3n-3 (*p* < 0.05) in rumen was higher for the C4 than for the C3 group. The differences between the FAs percentages in C3 and C4 rumen mostly mirrored those observed in the meat (Figure 2). As for C18:3n-3, its percentage was statistically higher in the C4 group than in the C3 one in the rumen compartment and numerically higher in the meat compartment (Figure 2). These results agree with those of Boerman et al. [23], who reported that the FAs amount available for absorption in the intestine was similar to that leaving the rumen.

The feed δ^13^C was different between the two experimental groups for all FAs, and this probably led to the differences observed in all the other compartments, the C4 group values being higher than those of the C3 one (*p* < 0.05, Figure 3). These results agree with many other studies [12,13,14]. The only exception, excluding those related to the lack of data for one or both diets, is represented by δ^13^C_C18:2n-6_MEAT_, which was not statistically different in the C3 and C4 diets (*p* > 0.05; Figure 3). Therefore, except for δ^13^C_C18:2n-6_MEAT,_ the δ^13^C of all FAs made it possible to discriminate, as for all compartments, between the cows belonging to the C3 and C4 groups (Figure 3). Finally, the low concentration of C18:3n-3 in meat did not make it possible to measure its δ^13^C.

### 2.2. FAs Quantification and Carbon Isotope Ratio Analysis: Compartment Effects

#### 2.2.1. FAs Quantification and Carbon Isotope Ratio Analysis in Feed and Rumen

In both groups, the content of C18:0 increased (*p* < 0.05), while that of C18:1n-9, C18:2n-2, and C18:3n-3 decreased (*p* < 0.05), passing from the diet to the rumen. Conversely, a different behavior was observed for C16:0 in the two groups, with a lower content in the rumen than in the diet in the C4 group, and vice versa in the C3 group (*p* < 0.05; Figure 1).

The FAs metabolic path starts with rumen. In this phase, dietary fats undergo chemical processes such as lipolysis, biohydrogenation, and de novo synthesis by ruminal microorganisms [24,25], which modify the profile of dietary FAs. The unsaturated FAs are mainly converted into C18:0, while a low percentage escape the biohydrogenation processes [21]. In particular, the disappearance of C18:3n-3 and C18:2n-6 in the rumen averages 93% and 85%, respectively [26]. These results agree with the data we obtained for the C18:2n-6 (loss of 97.6% in the C3 group and 98.7%in C4 group) and the C18:3n-3 (loss of 96.8% in the C3 group and 95.1% in C4 group). As the end product of the majority of the biohydrogenation processes, C18:0 has the highest amount of all FAs in the rumen (Figure 1), 16- and 8-fold higher in the rumen than in the diet for C3 and C4 diets, respectively.

Considering the C3 group, rumen had lower δ^13^C_C18:1n-9_ (*p* < 0.05) and higher δ^13^C_C18:3n-3_ (*p* < 0.05) than feed, with the δ^13^C of the other FAs not being statistically different in the two compartments. Conversely, rumen of C4 group had lower δ^13^C_C16:0_ (*p* < 0.05), δ^13^C_C18:1n-9_ (*p* < 0.05), δ^13^C_C18:2n-6_ (*p* < 0.05), δ^13^C_C18:3n-3_ (*p* < 0.05), but higher δ^13^C_C18:0_ (*p* < 0.05) than feed (Figure 3).

Around 20% of lipids leaving the rumen are of microbial origin, mainly from bacteria and protozoa [27]. Some types of the former use carbon sources for growth with very specific ^13^C signals or have a metabolism that results in characteristic isotopic ratios [28]. As the mean δ^13^C for all FAs in rumen ranges from −36.0‰ to −27.4‰ regardless of the diet, the possibility that the ruminal microbiota metabolism may result in products having this specific δ^13^C range of values cannot be excluded. The only exception to the mentioned range is C18:0 in the C4 group (δ^13^C_C18:0_RUMEN_C4_ = −24.3 ± 0.45‰), as its δ^13^C can be influenced by a wider range of factors, with C18:0 as the end product of the majority of the processes occurring in the rumen. As previously stated, its fate is linked to that of C18:1n-9, C18:2n-6, and C18:3n-3 through desaturation processes [21,27]. As these FAs are characterized by particularly high δ^13^C values in the C4 diet (Figure 3), their conversion in C18:0 may lead to a relatively high δ^13^C of this FA.

The framework becomes more complicated when considering that diets rich in fat [27] have a depressant effect on the biohydrogenation processes taking place in the rumen. In particular, 10% of fat added into the diet leads to a reduction of fermentation by 50%, especially if unsaturated FAs are supplemented [27]. This may have to be considered, to some extent, in the case of the C4 diet, which is 1.5-fold higher in fat content with respect to the C3 one.

Furthermore, it is well known that corn silage has a much smaller particle size than hay and this could lead to a lower retention time of the feed into the rumen [29]. This factor may also significantly affect lipid metabolism by causing differences in the digestive processes and promoting the selective growth of some types of bacteria [30]. This could contribute to widening the different compartments effect between C3 and C4 groups as for the FAs profile and their δ^13^C.

#### 2.2.2. FAs Quantification and Carbon Isotope Ratio Analysis in the Liver

The content of all FAs increased significantly (*p* < 0.05) in the liver, as against the rumen, for both groups (Figure 1). Following the digestive process, the FAs reach the liver and are carried by blood. Its flow and the FAs concentration influence their supply to this organ, which can therefore be seen as the next step after the rumen in the FAs metabolic path [31]. As the liver of ruminants has a low capacity for the de novo FAs synthesis [32], here, FAs are more likely to be β-oxidised to produce free energy or esterified to triglycerides (TAG) [33]. Moreover, in the liver of ruminants, exogenous FA are preferentially stored as TAG in lipid droplets, while de novo synthesized FA are likely channeled to TAG synthesis for very low-density lipoproteins formation [34]. Furthermore, the nature of the diet (fat to carbohydrate ratio, fat content, FAs composition) regulates the partitioning of fats between the various hepatic metabolic pathways. For example, insulin stimulates de novo lipogenesis and FA esterification [35] but inhibits FA oxidation [36].

The δ^13^C of all FAs increases (mostly being statistically different) going from the rumen to the liver for both diets (Figure 3). This may be due to the fact that the path in between the two compartments also entails chemical reactions, as previously mentioned, leading to fractionation processes. Indeed, depending on their origin (whether de novo synthesized or exogenous), the FAs have different fates in the liver: the former are likely channeled to triglycerides synthesis for very low-density lipoproteins formation, while the latter are preferentially stored as triglycerides in lipid droplets [37]. Moreover, the diet type (in terms of fat content and FAs composition, especially) and hormones regulate the partitioning of fats between the various hepatic metabolic pathways [32].

#### 2.2.3. FAs Quantification and Carbon Isotope Ratio Analysis in the Meat

The adipose tissue can be considered as one of the final steps of FAs metabolism. The FAs of the meat may derive either from the diet only, as in the case of essential linoleic and linolenic FA, or from de novo endogenous synthesis, or both [1]. In the adipose tissue, which accounts for >90% of the de novo FAs synthesis in non-lactating ruminants [38], lipolysis and FAs esterification proceed continuously [39].

The amount of C16:0, C18:0, and C18:1n-9 increased significantly (*p* < 0.05) in the meat with respect to the rumen, regardless of the experimental group (*p* < 0.05; Figure 1). In this compartment, C16:0 is subjected to de novo synthesis and is a precursor of C18:0 production through elongation [40,41]. Moreover, C18:0 is partially converted into C18:1n-9 through the activity of delta-9 desaturase [40], which results evident from the FAs percentages in Figure 2. Indeed, the C18:0 percentage in meat is lower than the value observed in the rumen (*p* < 0.05), while the highest value of C18:1n-9 was found in the meat (*p* < 0.05; Figure 2). As expected, for both C3 and C4 groups, the contents in C18:2n-6 and C18:3n-3 are lower (*p* < 0.05) in meat than in the diet, as these FAs have a dietary origin and are partially subject to rumen biohydrogenation (Figure 1).

As shown in Figure 3, the highest δ^13^C of all FAs of the C3 group was found in meat (*p* < 0.05). On the other hand, in the C4 group meat compartment, only δ^13^C_C18:1n-9_ resulted to be higher than the same value observed in the rumen (*p* < 0.05). Furthermore, all the other FAs of the C4 group had similar δ^13^C in meat and rumen (*p* > 0.05). Considering that the enrichment of δ^13^C is also related to the de novo synthesis [42], it is possible to hypothesize that the portion of FAs of exogenous origin and the portion of FAs deriving from the endogenous synthesis in meat may be different in the C3 and C4 groups. Nevertheless, further studies should be carried out in order to validate this assumption.

## 3. Materials and Methods

The study met the EU Directive 2010/63/EU. All the procedures were non-invasive and routine. Even though formal approval was not required, the ethical committee of the University of Udine approved the trial (Prot. No.8/2018).

### 3.1. Description of the Samples

#### 3.1.1. Diet

Thirteen Italian Simmental multiparous cull cows were assigned, on the basis of their body weight (601 ± 21.4 kg; *p* > 0.05), age (83 ± 9.8 months; *p* > 0.05) and body condition score (3.1 ± 0.14 points; *p* > 0.05, [43]), to two dietary treatments, either C3 (C3 group; *n* = 6) or C4 (C4 group, *n* = 7). The groups differed in the metabolism of the plants the diet was based on, as previously mentioned. The C3 group was offered hay *ad libitum* and received a concentrate (6.9 kg dry matter (DM)) made up of wheat meal, barley meal, wheat bran, hempseed cake, soybean meal, mineral, and vitamins. The hay was composed of 70.1% DM neutral detergent fiber (NDF), 7.1% DM crude protein (CP), and had net energy for lactation (NEL) of 4.3 MJ/kg of DM. The concentrate was made up of 13.9% DM CP and 7.8 MJ NEL/kg DM. The C4 group was offered corn silage *ad libitum* and received 3.2 kg DM of a concentrate made up of 84% corn meal and corn gluten (C4 plants), while C3 plants (soybean meal and hempseed cake) represented the remaining part. The corn silage was composed of 7.8% DM CP, 42.8% DM NDF, and 5.9 MJ/kg of DM NEL, while the concentrate had 23.4% DM CP and 7.7 MJ NEL/kg DM. The diets were formulated hypothesizing a target daily weight gain of 1.1 kg/d for all the animals according to INRA standards [44]. The concentrates were individually offered in twice-daily meals and were completely consumed by the animals. During the entire experimental period (4 months), the animals stayed healthy, as verified by periodical clinical examinations by the veterinary.

To ensure the individual daily forage (hay and silage) intake, all cows were equipped with a noseband pressure sensor (RumiWatch system, ITIN-HOCH GmbH, Liestal, Switzerland) to evaluate the feeding behaviors, including eating time as well as chews and number of boluses [45,46]. In particular, the relationship between forage intake and feeding behavior had been studied in a preliminary period involving the same animals and forage (hay and silage) batches similar to those used during the experimental period [47]. During the trial, the mean DM intake of hay and silage were 9.3 ± 1.0 kg of DM and 13.0 ± 1.2 kg of DM, respectively. The fat content of C3 and C4 diets was 25.3 and 38.5 g/kg DM, respectively. The dietary plants in the two groups were classified as 100% C3 (C3 group) and as over 95% C4 (C4 group). Samples of hay, silage, and concentrates were collected every two weeks, dried at 65 °C in a forced draft oven for 48 h, and used for stable isotopes and FAs analyses. The individual intake of hay, silage, and concentrates was considered to calculate the average δ^13^C and FAs amount of the diets.

#### 3.1.2. Slaughter

The day before slaughter, the animals accessed the morning meal only. The cows were slaughtered at the same live weight for both experimental groups (703 ± 23.2 kg; *p* > 0.05), in an EU-licensed abattoir (40 km far from the farm) within 20 min from their arrival. At slaughter, individual samples of the rumen (≅100 g of representative content), liver (≅100 g) and meat (≅100 g of m. longissimus thoracis at 6–7 rib level) were immediately collected and refrigerated. All samples were freeze-dried (Lyo Quest, Telstar, Legnago, Italy) and ground in a laboratory blender for stable isotope and fatty acids analyses. The pH of the rumen fluid was measured by using a pH meter (Hanna, HI 8424, Padova, Italy) equipped with a glass electrode (Crison, 5232, Barcelona, Spain).

### 3.2. Extraction, Derivatization and Quantification of the FAs

#### 3.2.1. Lipid Extraction

Lipid extraction was carried out through Accelerated Solvent Extraction (ASE 350 Dionex, Thermo Fisher Scientific, Rovigo, Italy), using a mixture of chloroform: methanol (2:1, *v*/*v*) [48]. Samples were placed in 22 mL extraction cells, in variable amounts (1.0–2.5 g) according to the expected fat content. A weighted amount of C21:0 (or C19:0 in the case of forage samples) was added as a standard for lipid determination. The sample loading was completed by filling with Diatomeehydromatrix (Agilent Technologies, Santa Clara, CA, USA) and sealing off each extraction cell with a cellulose filter. After the extraction, the solvent was evaporated with Univapo 100 ECH first (Vacuum concentrator centrifuge, Elettrofor, scientific instrument, Rovigo, Italy) and then with a N_2_stream to constant weight. The samples were finally stored in hexane to carry out the derivatization.

#### 3.2.2. Derivatization

The fatty acid methyl esters (FAMEs) were obtained through the transesterification of triglycerides by using the method described by Perini et al. [49]. The hexane extract (2 mL) was put in a reactor, where the solvent was evaporated under a stream of N_2_. Thus, 1.5 mL of a mixture of sulphuric acid/methanol (1:16, *v*/*v*) was added. The sample was heated at 100 °C for 2 h and shaken during the heating. A water solution NaCl saturated (5 mL), followed by 3 mL of hexane, was added to the cooled solution, which was therefore vigorously shaken. Once the solution had stratified in two layers, the upper organic phase was picked up with a micropipette and injected into the GC-MS and GC-C-IRMS.

#### 3.2.3. FAs Quantification by GC-MS

The GC-MS analyses were carried out in EI mode (70 eV) with a 5977E MSD system, a single-quadrupole with hyperbolic rod sensors (Agilent Technologies, Santa Clara, CA, USA), a 7820A GC system equipped with a 7693A autosampler and an automatic split/splitless injector. The samples (1 µL) were injected in split mode (1:50) and the separation was performed on an HP-88 fused silica capillary column (100 m, 0.25 mm ID, 0.20 µm film thickness, 88%-Cyanopropy-aryl-polysiloxane) (Supelco, Bellefonte, PA, USA). The GC oven was set on the following temperature program: 50 °C (hold 5 min); ramp (8 °C/min) to 160 °C; 160 °C (hold 4 min); ramp (0.6 °C/min) to 200 °C; 200 °C (hold 1 min); ramp (15 °C/min) to 240 °C; 240 °C (hold 5 min). Helium (6.0) was used as carrier gas with a constant flow rate (1.2 mL/min). Temperature values of the ion source, the quadrupole, and the transfer line were set to 240 °C, 150 °C, and 230 °C, respectively. The GC-MS analyses were performed in full scan mode (*m*/*z* 50–600) after a solvent delay of 15 min, with 3 microscan/s.

### 3.3. Isotopic Analysis of FAs by GC-IRMS

The isotopic analysis of carbon on the individual FAMEs was performed by injecting the samples (1 μL) in split mode (1:10 with a split flow 20 mL/min) through an autosampler (Triplus, Thermo Scientific, Bremen, Germany) into a GC (Trace GC Ultra, Thermo Scientific, Bremen, Germany). After the GC column, the flow was split into two parts, 1/10 of the flow was sent to a conventional MS, while the remaining part (9/10 of the flow) was directed to an IRMS. The GC was connected in parallel with a single-quadrupole MS (ISQ Thermo Scientific, Bremen, Germany) to identify each compound and with a Thermo Scientific GC IsoLink II Conversion Unit, interfaced to a Thermo Scientific DELTA V Isotope Ratio Mass Spectrometer via a Thermo Scientific ConFlo IV.

The separation of the FAs was performed on a BPX-70 polar fused silica capillary column (60 m, 0.32 mm ID, 0.25 µm film thickness) supplied by Trajan Scientific and Medical (Victoria, Australia). The injector temperature was set at 250 °C. The GC oven was set on the following temperature program: hold 4 min at 50 °C; ramp with a speed of 30 °C/min to 170 °C; ramp with a speed of 2 °C/min to 200 °C; ramp with a speed of 1 °C/min to 210 °C; hold 8 min at 210 °C. Helium (5.0) was used as a carrier gas, having a constant flow rate of 2.0 mL/min. The FAs were identified by a single-quadrupole GC-MS (ISQ Thermo Scientific, Bremen), directly connected with the GC. The ion source and the transfer line were both set at 250 °C. The GC/MS analyses were performed in full scan mode (*m*/*z* 35–600) after a solvent delay of 10 min. Each compound was identified by comparing the mass spectra, acquired at 70 eV, with the NIST library data (NIST Mass Spectral Search Program (Version 2.0f) and NIST Standard Reference Database 1A NIST/EPA/NIH Mass Spectral Library (NIST 08)).

To determine δ^13^C, the compounds exiting the GC were combusted into CO_2_ and H_2_O in a combustion furnace reactor (GC IsoLink II Conversion Unit), which was set at a temperature of 1030 °C and consisted of a non-porous alumina tube (320 mm) containing three wires (Ni/Cu/Pt, 0.125 mm of diameter, 240 mm long) braided and centered end-to-end within the tube. Water was subsequently removed thanks to a Nafion dryer (integrated in the IsoLink) before the gaseous analytes were transferred to the IRMS.

#### Results Elaboration

The δ^13^C values were calculated against international reference materials, injected at the beginning and the end of each analytical sequence (Eicosanoic Acid Methyl Esters USGS70 (δ^13^C = −30.5‰) and USGS71 (δ^13^C = −10.5‰) (United States Geological Survey, Reston, VA, USA)). The values were expressed in the delta in relation to the international V-PDB (Vienna-Pee Dee Belemnite), according to the following general equation:$$\delta_{ref}{(}^{i}E{/}^{j}E,sample)=\left[\frac{R{(}^{i}E{/}^{j}E,\;\;\;sample)}{R{(}^{i}E{/}^{j}E,\;\;\;ref)}\right]-1$$ where *ref* is the international measurement standard which, by definition, forms the zero point of the isotope delta scale, sample is the analysed sample and *^i^E*/*^j^E* is the isotope ratio between heavier and lighter isotopes. The resulting delta values were multiplied by 1000 and expressed in units “per mil” (‰). Each sample was analyzed in triplicate. Measurement uncertainty (2S_R_) was calculated as < 0.5‰ as for δ^13^C analysis.

Finally, it must be considered that the δ^13^C value of FAMEs derives from the δ^13^C of the FAs and from the contribution of the δ^13^C of the reagent (methanol in the present study) used for the derivatization reaction. An empirical correction was therefore applied to determine the real δ^13^C value of the FAs by applying the following equation:
(C_n_ + 1) δ^13^C _FAME_ = C_n_ δ^13^C _FA_ + δ^13^C _Me_
where δ^13^C_FAME_, δ^13^C_FA,_ and δ^13^C_Me_ are the carbon isotopic values of the FAME, the FA, and of the methanol, respectively. The C_n_ corresponds to the number of carbons in the FA and the δ^13^C of the Me (−35.8‰ ± 0.1‰) was determined using EA-IRMS (Elemental Analyser—Isotope Ratio Mass Spectrometer) (Flash 1112, Thermo Scientific, Bremen, Germany).

### 3.4. Statistical Analysis

The statistical analysis was performed using R software, vers. 4.0.4 (R Foundation for Statistical Computing, Vienna, Austria) and SPSS vers. 17 software (SPSS Inc., Chicago, IL, USA). Variables were analyzed with a mixed model for repeated measures that considered compartment (feed, rumen, liver, and meat) and diet (C3, C4) as within and between subject factors, respectively. The compartment × diet interaction was also considered, and the analyses were carried out as suggested by Wang and Goonewardene [50]. The normality of the data distribution was assessed using Shapiro–Wilk test and the normal Q–Q plot of residuals. When appropriate, data were transformed for parametric testing and, if the model assumptions were not yet satisfied, generalized estimating equations applied on the same model were considered. The Bonferroni adjustments were used for multiple comparisons. A *p*-value < 0.05 indicates significant statistical differences.

## 4. Conclusions

In this study, 13 multiparous cull cows were considered. The quantification through GC-MS and the carbon stable isotope analysis through GC-C-IRMS of five fatty acids are provided. The analyses were carried out on samples of four compartments (feed, rumen, liver, meat). Two different diets, based on C3 or C4 plants, were considered for each compartment. The isotopic results made it possible to discriminate between different diets in all the compartments, with the only exception of C18:2n-6 in the meat. The δ^13^C analysis and the quantification of the FAs helped in shedding light on the metabolic path that the FAs follow in the bovine organism and on how this path changes depending on the diet. Nevertheless, the metabolic path of the FAs is still far from being completely understood, thus more studies, especially on the various chemical reactions taking place in the rumen, should be carried out.

## Figures and Tables

**Figure 1 molecules-27-02310-f001:**
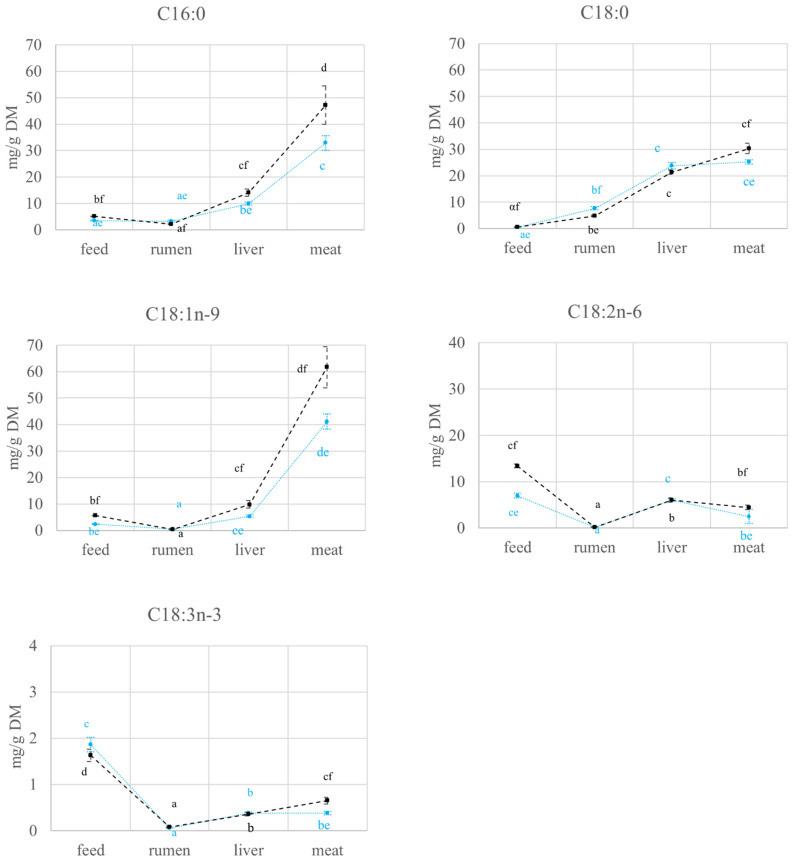
Variations (mean ± se) of FAs expressed as mg/g DM (dry matter) in compartments (feed, rumen, liver, meat) of cull cows fed with diets based on C3 (C3 group, blue) and C4 (C4 group, black) plants. a, b, c, d: *p* < 0.05 within experimental group; e, f: *p* < 0.05 within compartment.

**Figure 2 molecules-27-02310-f002:**
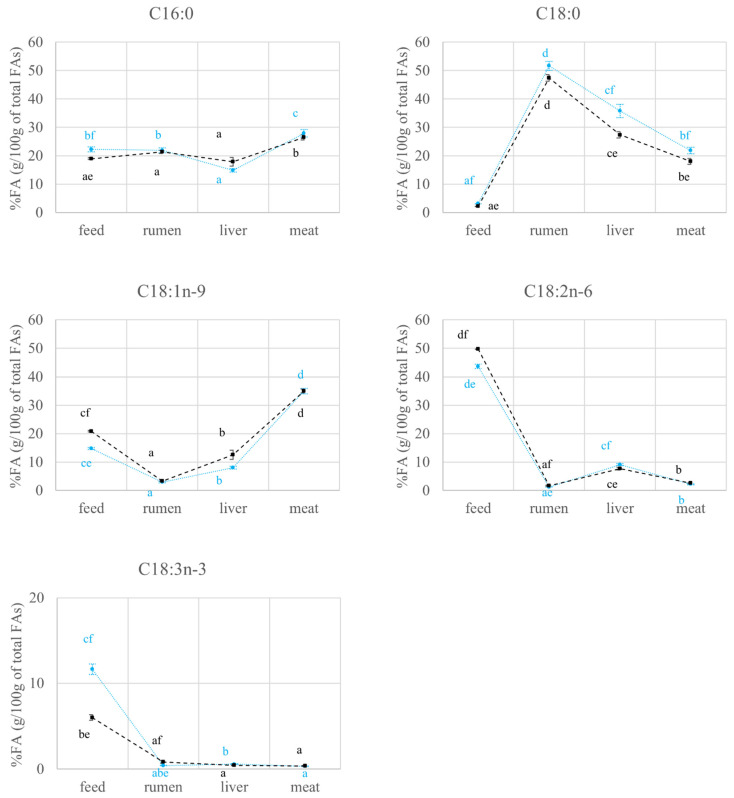
Variations (mean ± se) of FAs expressed as a percentage (g/100 g of total FAs) in various compartments (feed, rumen, liver, meat) of cull cows fed with diets based on C3 (C3 group, blue) and C4 (C4 group, black) plants. a, b, c, d: *p* < 0.05 within experimental group; e, f: *p* < 0.05 within compartment.

**Figure 3 molecules-27-02310-f003:**
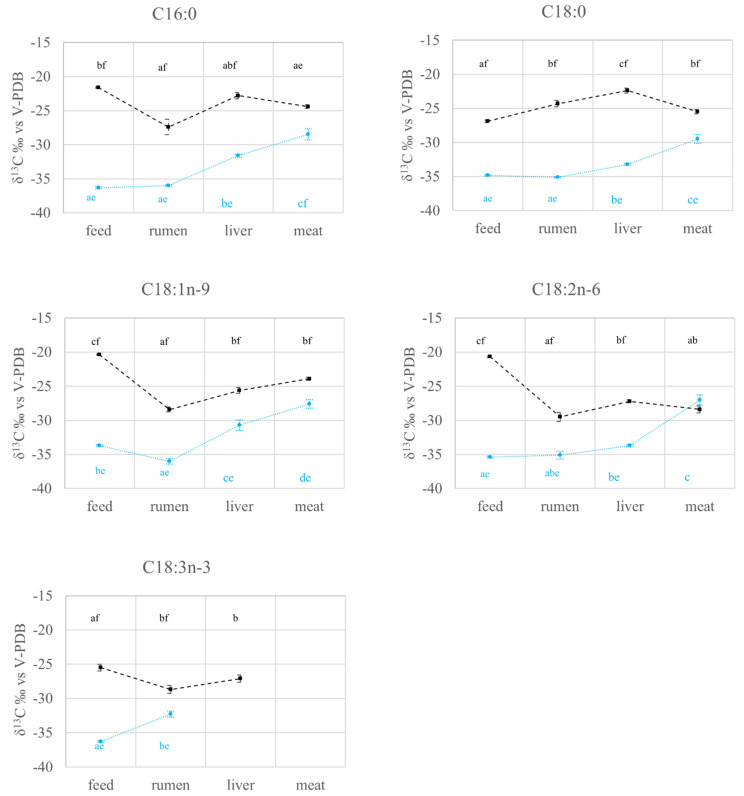
Variations (mean ± se) of δ^13^C (‰) of FAs in different compartments (feed, rumen, liver, meat) of cull cows fed with diets based on C3 (C3 group, blue) and C4 (C4 group, black) plants. a, b, c: *p* < 0.05 within experimental group; e, f: *p* < 0.05 within compartment.

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
