# Peer review of "Gas Chromatography Combustion Isotope Ratio Mass Spectrometry to Detect Differences in Four Compartments of Simmental Cows Fed on C3 and C4 Diets"

_molecules, 2022, doi:10.3390/molecules27072310_

Round 1

Reviewer 1 Report

The MS of high quality and  deseve publication  after minor revsion inculding the following points:

1- The novelty/add value of the work should be add to the introduction section.

2-References should  be updated to 2022 if any.

3- L 216-228, I suggest to make a table for diet profile and nutrietnts composition of the tested diets.

4- The conculsion section could be reduced by 50% with concnetration on the main outputs of the results and its application in beef cattle nutritions.

5-References 1; 4; 11; 16, 21; 22; 24; 25; 27; 34,& 38 the year of publctaion should be bolded, Plz check others. 

Author Response

Please, see the attached PDF.

Reviewer 2 Report

Dear Authors,

The present manuscript “Gas chromatography combustion isotope ratio mass spectrometry to detect differences in four compartments of Simmental cows fed on C3 and C4 diets” [Manuscript ID: molecules-1634421] is the important and actual work; it is in a frame of the journal scope. I do not doubt the technical quality of the work and feel that there is a sufficient impact on a broader readership to justify publication in the "Molecules".

This article described five major fatty acids (FAs): linoleic and linolenic acids (“essential”), as well as palmitic, stearic and oleic acids (“non-essential”) in four compartments (feed, rumen, liver, meat) of animals fed two different diets (based on either C3 or C4 plants).

Comments: 

  1. An important correction must be done in the “Abstract”. The authors wrote: “The aim of this study was to assess the possibility to discriminate the diets of cows in different animal compartments and to study the fate of dietary FAs in the bovine organism, using carbon isotopic ratios”. That is why the sentence “Different trends in the δ13C throughout the compartments of the C3 and C4 groups suggested different FAs fates depending on the diet” must be enlarged in a way to describe “the fate of dietary FAs” in the liver and meat. It is clear that the abstract “should be a total of about 200 words maximum”. But there is some space: 1) the abstract presented by the authors is about 174 words; 2) the last sentence in the abstract is not very informative and can be decreased by 15 words (i.e. cut the part = “….and can provide information about the metabolic path of the FAs in the bovine organism”) that gives more place to describe “the fate of dietary FAs” in the liver and meat.
  2. The major corrections must be done in the parts 2.2.2. and 2.2.3. It is important to describe in more details or show a scheme of the proposed “FA metabolic pathways” as additional Figures in the parts 2.2.2. and 2.2.3. In particular:

2A) The text at lines 177-185 in the part “2.2.2. FAs quantification and carbon isotope ratio analysis in the liver” must be enlarged in order to describe in all possible details (even in the terms “highly likely” etc.) “the fate of dietary FAs” in the liver.

2B) The text at lines 198-205 in the part “2.2.3. FAs quantification and carbon isotope ratio analysis in the meat” must be enlarged in order to describe in all possible details (even in the terms “highly likely” etc.) “the fate of dietary FAs” in the meat.

  1. Please, check some misprints.

I strongly recommend to accept the present manuscript after minor revision.

Author Response

Please, see the attached PDF.
